

# OMI Satellite Observations of decadal changes in Ground-Level Sulfur Dioxide over North America

Shailesh K. Kharol[1], Chris A. McLinden[1], Christopher E. Sioris[1], Mark W. Shephard[1], Vitali Fioletov[1], Aaron van Donkelaar[2], Sajeev Philip[2], Randall V. Martin[2]

[1]Air Quality Research Division, Environment and Climate Change Canada, Toronto, Ontario M3H 5T4, Canada
[2] Department of Physics and Atmospheric Science, Dalhousie University, Halifax, Nova Scotia, Canada

*Correspondence to*: S. K. Kharol (shailesh.kharol@canada.ca)

**Abstract.** Sulfur dioxide ($SO_2$) has a significant impact on the environment and human health. We estimated ground-level sulfur dioxide ($SO_2$) concentrations from the Ozone Monitoring Instrument (OMI) using $SO_2$ profiles from the Global

Environmental Multi-scale – Modelling Air quality and CHemistry (GEM-MACH) model over North America for the period of 2005-2015. OMI-derived ground-level $SO_2$ concentrations (r = 0.61) and trends (r=0.74) correlated well with coincident in-situ measurements from air quality networks over North America. We found a strong decreasing trend in coincidently sampled ground-level $SO_2$ from OMI (-81±19%) and in-situ measurements (-86±13%) over Eastern US for the period of 2005-2015, which reflects the implementation of stricter pollution control laws including flue-gas desulfurization (FGD)

devices in power plants. The spatially and temporally contiguous OMI derived ground-level $SO_2$ concentrations can be used to assess the impact of long-term exposure to $SO_2$ on the health of humans and the environment.

## 1 Introduction

Sulfur dioxide ($SO_2$) is a short-lived atmospheric trace gas emitted into the atmosphere from natural (e.g. volcanic eruption, oxidation of dimethylsulphate (DMS) over oceans) and anthropogenic sources (e.g. combustion of fossil fuels and smelting

of sulfur-containing metal ores), and plays a pivotal role in the global sulfur cycle. $SO_2$ has a short lifetime of hours to days, and it oxidizes quickly in the atmosphere to produce sulfate aerosols that affect the climate (Intergovernmental Panel on Climate Change, 2013) and the environment from local to regional and global scales. Sulfate aerosols are major contributor to $PM_{2.5}$ (particulate matter with aerodynamic diameter < 2.5 μm) chemical composition and accounts for 17% and ~30% of the annual mean $PM_{2.5}$ mass globally and over eastern United States (Philip et al., 2014). Sulfate aerosol formation leads to

degradation in visibility and air quality (van Donkelaar et al., 2008), deposition of sulfuric acid (Dentener, et al., 2006; Vet et al., 2014), and poses a serious health hazard to the general population (Lee et al., 2015). The increased risk of premature mortality associated with $SO_2$ alone or its secondary pollutants has been emphasized in several epidemiological studies (Chinn et al., 1981; Derriennic et al., 1989; Hatzakis et al., 1986; Krzyzanowsli & Wojtymiak, 1982). Furthermore, it has been recently reported by Lelieveld et al., (2015) using the EMAC (ECHAM5/MESSy Atmospheric Chemistry) general

circulation model that in the U. S., in addition to agricultural emissions (an important source of ammonia ($NH_3$)), emission



from coal fired power plants (an important source of $SO_2$ and nitrogen oxides ($NO_x$)) is the largest contributor to premature mortality in 2010. Due to the adverse impact on the environment and human health, $SO_2$ and its oxidation products (i.e. fine particulate matter ($PM_{2.5}$)) are considered as designated criteria pollutants in European Union (European Commission, http://ec.europa.eu/environment/air/quality/standards.htm), United States of America (US Environmental Protection Agency
(EPA), https://www.epa.gov/criteria-air-pollutants) and Canada (https://www.ec.gc.ca/Air/default.asp?lang=En&n=7C43740B-1).

Globally, atmospheric $SO_2$ is monitored regularly through a relatively small number of measurement networks, that produce accurate measurements, but over a limited spatial area. Satellite measurements have the advantage of providing complete
daily global coverage of $SO_2$. Satellite observations of $SO_2$ vertical column density begun in the 1980s but the launch of the Ozone Monitoring Instrument (OMI) (Krotkov et al., 2006, Yang et al., 2007) on the Aura satellite in 2004 has enabled large point sources to be resolved with its higher spatial resolution (13 x 24 $km^2$ at nadir) (Fioletov et al., 2013). Satellite measurements of $SO_2$ have been used to identify and analyze emissions (Fioletov et al., 2011, 2013, 2015; Lee et al., 2011; McLinden et al., 2016a), track changes in total column density (McLinden et al., 2016b; Krotkov et al., 2016) and estimate
dry deposition flux (Nowlan et al., 2014). However spatiotemporal variations in ground-level $SO_2$ have not yet been assessed from the satellite observations. Recently, a decreasing trend in $SO_2$ emissions and particulate sulfate has been reported by Hand et al. (2012) over the United States from the early 1990s through 2010.

In this paper we first describe the OMI $SO_2$ product, in-situ measurement network, GEM-MACH model, ground-based $SO_2$
estimation from OMI and trend analysis. We then use this data and methodology to estimate ground-level $SO_2$ from OMI and evaluate it with coincident in-situ measurements over North America for the period of 2005-2015. These results are then used to determine the trend in ground-level $SO_2$ from both OMI and collocated in-situ measurements.

## 2 Data sets & methodology

### 2.1 OMI

OMI is a nadir-viewing UV-visible spectrometer boarded on the Aura satellite that was launched in July 2004, and is part of the NASA A-train constellation (Levelt et al., 2006). The Aura satellite overpasses the equator at early afternoon (1300-1430 local time) in sun-synchronous ascending polar orbit. OMI provides daily global coverage of aerosols and trace gases, including $SO_2$, with a variable ground spatial resolution of 13 km $\times$ 24 km at nadir to 140 km $\times$ 26 km at swath edge. We use the OMI operational Principal Component Analysis (PCA) $SO_2$ product (OMSO2 v1.2.0), which is publically available from
NASA Goddard Earth Sciences (GES) Data and Information Services Center (DISC) (http://disc.sci.gsfc.nasa.gov/Aura/data-holdings/OMI/omso2_v003.shtml). The details of PCA algorithm can be found elsewhere (Li et al., 2013). In brief, this algorithm applies the PCA technique to OMI-measured radiances between 310.5 and





340 nm to extract principal components from each row on an orbital basis. The PCA algorithm replaced the Band Residual Difference algorithm (Krotkov et al., 2006) as the operational algorithm for the standard OMI $SO_2$ data because only the latter algorithm was biased (Fioletov et al., 2013; Krotkov et al., 2016). Also, $SO_2$ variability is reduced by a factor of two in the PCA algorithm (Li et al., 2013). Here, we exclude the cross-track pixels affected by row anomaly

(http://www.knmi.nl/omi/research/product/rowanomaly-background.php), which was first noticed in the data in June 2007. We use OMI $SO_2$ columns with cloud radiance fraction <0.2, and solar zenith angles <65°. We exclude from the analysis the OMI $SO_2$ data affected by the largest northern mid-latitude volcanic eruptions in the OMI time frame namely Kasatochi (Aleutian Islands, Alaska, August 2008, 52°N) and Sarychev (Kuril Islands, Eastern Russia, June 2009, 48°N). Here, we used the mean OMI values over a 32 km averaging-radius (Fioletov et al., 2011) that is oversampled onto a 0.1° x 0.1°

latitude/longitude grid.

## 2.2 $SO_2$ monitoring networks

To evaluate the OMI-derived ground-level $SO_2$ we use hourly in-situ $SO_2$ measurements from the Air Quality System (AQS) network of the US EPA (http://www.epa.gov/ttn/airs/airsaqs/detaildata/downloadaqsdata.htm) and Environment and Climate Change Canada's National Air Pollution Surveillance (NAPS) network (http://maps-cartes.ec.gc.ca/rnspa-naps/data.aspx)

over the US and Canada for the period of 2005 to 2015. US-EPA AQS site locations vary from regional background to urban and industrial locations, and measures $SO_2$ using continuous gas monitors. The Canadian NAPS sites are generally located in populated areas. The hourly in-situ measurements are averaged over a 2 h period (1300-1500 Local Time) to correspond with the satellite observation times over North America.

## 2.3 Model information

We use the Global Environmental Multi-scale – Modelling Air quality and CHemistry (GEM-MACH) model for the tropospheric $SO_2$ profile to relate the OMI $SO_2$ column to ground-level concentrations. GEM-MACH is the Canadian regional air quality forecast model used operationally to predict the concentrations of $O_3$, $NO_2$, and $PM_{2.5}$ over North America (Moran et al., 2010, Gong et al., 2015). GEM-MACH model utilizes emissions inventories from US EPA and Environment Canada data for the year 2006. It uses detailed tropospheric processes for gas and particle chemistry and

microphysics originating in the offline AURAMS model (A Unified Regional Air-quality Modelling System; Gong et al., 2006), and incorporates them on-line into the Canadian weather forecast model (Global Environmental Multiscale model, Côté et al., 1998). A detailed description of the chemical processes found in AURAMS and GEM-MACH is provided elsewhere (Kelly et al., 2012). The results used here are from archived forecasts from 2010 to 2011 for a domain covering North America at 15 km×15 km resolution. The lowest model layer is taken as ground-level concentration. More details on

Environment Canada Air Mass Factors calculation for $SO_2$ are discussed in McLinden et al., 2014; 2016b.





## 2.4 Estimation of ground-level SO$_2$ from OMI

The ground-level SO$_2$ mixing ratio from OMI is estimated using the approach described by Lamsal et al. (2008) over North America for the period of 2005-2015. The ground-level SO$_2$ mixing ratio $S$ is estimated from the local OMI tropospheric SO$_2$ column $\Omega$ as:

$$S_{OMI} = \Omega_{OMI} \times \frac{S_{model}}{\Omega_{model}} \tag{1}$$

The subscript $model$ represents GEM-MACH model. More details on the procedure are discussed in McLinden et al. (2014).

## 2.5 Trend Analysis

We analyzed the trends in monthly ground-level SO$_2$ over North America from OMI and in-situ measurements for the period of January 2005 - December 2015. We applied a general least squares regression following Boys et al. (2014) and Kharol et

al. (2015) using the basic model:

$$x = z\beta + e, \quad e \sim N(0, \sigma^2 V) \tag{2}$$

where, for a time series of n months, $x$ is a time series vector (n × 1) containing SO$_2$ surface mixing ratio values; $z$ is a
design matrix (n × 2) for the linear model; $\beta$ is a vector (2 × 1) containing the intercept and slope of the linear model; $e$ is an error vector (n × 1) containing the residuals which, for validity, should be approximately normally distributed with zero mean, however, is permitted to covary with adjacent values according to $V$ – a positive definite, symmetric covariance matrix, to accommodate possible autocorrelation between adjacent months. Correlated errors between adjacent months are represented by a first order autoregressive model of $e$, which can be expressed as:

$$e_t = \emptyset e_{t-1} + w_t \qquad t = 1, \dots n, \; w \sim N(0, \sigma^2 I) \tag{3}$$

where, the residual $e_t$ for month $t$ is a fraction $\emptyset$ of the previous month's residual $e_{t-1}$ with a white noise component $w_t$ which, for validity, should be approximately normally distributed with zero mean, constant variance and independent. We
deseasonalized the monthly time series by subtracting the climatological monthly median prior to regression. Note, the trend is more heavily weighted toward summer when observations are more frequent.

## 3 Results & Discussion

Figure 1 shows the spatial distribution of mean OMI derived ground-level SO$_2$ over North America for the periods of 2005-2007, 2008-2010, 2011-2015 and 2005-2015. The major SO$_2$ hot-spots (that is, locations of high SO$_2$ associated with a large
nearby source) are primarily located in the Eastern US from coal-fired power plants and industrial activities (Krotkov et al.,



2016). There are far fewer sources in the western US and Canada, with a few notable exceptions such as Flin Flon, Snow Lake, Sudbury, Thompson, Montreal, the oil sands region in northern Alberta and power plants nearby Edmonton. The spatial distribution of annual mean OMI derived ground-level $SO_2$ for each year is shown in supporting information Figure S1. A noticeable decrease in OMI derived ground-level $SO_2$ is apparent from Figure 1 during 2008-2010 and 2011-2015 compared to 2005-2007. These US reductions correspond with the installation of flue-gas desulfurization (FGD) units at many power plants to meet stricter emissions limits introduced by the Clean Air Interstate Rule. The closure of Flin Flon (54.76$^o$ N, 101.87$^o$ W) copper smelter is also apparent. The OMI derived ground-level $SO_2$ concentrations over large geographical area could be useful to assess its impact on human health and environment. It can also provide valuable information to policy makers where air quality network measurements are not available.

To verify these satellite findings, we compared the OMI-derived ground-level $SO_2$ concentrations with in-situ measurements over North America for the period of 2005-2015. The original OMI-derived ground-level $SO_2$ concentration (black circles) moderately correlates with collocated in-situ measurements (r = 0.61), but have a significant difference in slope (slope = 0.39) (Fig 2). The departure from unity of the slope is a common feature of virtually all satellite-surface comparisons of this kind (Kharol et al., 2015), and can be a result of both the in-situ monitor placements (i.e. mainly located in the cities and close to pollution sources) and differences in the spatial sampling of the two types of observations. To quantify this inhomogeneity effect we utilized output from the GEM-MACH model at high-resolution (2.5 km x 2.5 km; supporting information Fig S2) over a region in central Canada. These high-resolution GEM-MACH $SO_2$ columns at the locations of the in-situ monitors were taken as representative of point (in-situ) measurements. The model $SO_2$ columns were then progressively averaged up (smoothed) to 30 km x 30 km, approximately representing the spatial size of an OMI pixel. The smoothed columns are regressed against the unsmoothed columns. The slope and correlation coefficient continue to decrease from unity as the smoothing is increased. We used this estimate of the spatial inhomogeneous sampling obtained from the original (2.5 km) vs smoothed (30 km) GEM-MACH $SO_2$ column (supporting information Fig S3) to derive a scaling factor (in-situ scaled = 0.52 × (in-situ) + 0.04, R = 0.83) that is used to adjust the in-situ measurements to be representative of the OMI pixel size over all of North America. We noticed ~92% increase in slope to 0.75 when comparing the spatial inhomogeneity adjusted in-situ measurements with the OMI ground-level $SO_2$ (red circles in Fig. 2). In comparison to previous studies, Lee et al., (2011) comparing ground-level $SO_2$ mixing ratios derived from SCIAMACHY and OMI with in-situ measurements from US-EPA AQS and NAPS monitoring networks over the United States and Canada for the year of 2006 reported slightly higher correlation (r = 0.86, slope = 0.91 for SCIAMACHY and r = 0.80, slope = 0.79 for OMI). In their study they used 15 km coincidence criterion and included only AQS sites measuring less than 6 ppbv at satellite overpass times. Nowlan et al., (2011) estimated ground-level $SO_2$ from GOME-2 and compared with in-situ measurements over North America from Clear Air Status and Trends Network (CASTNET; r = 0.85) and US-EPA AQS and NAPS (r = 0.40) for 2008.





We determined the trend in ground-level $SO_2$ from OMI using the monthly time series from January, 2005 to December, 2015. Figure 3 illustrates the spatial distribution of OMI-derived ground-level $SO_2$ trend over North America for the period of 2005-2015. We noticed a strong decreasing trend in ground-level $SO_2$ over eastern US and Flin Flon in Canada. The observed decrease in ground-level $SO_2$ concentration in the eastern US corresponds to more strict pollution control laws

implemented to reduce $SO_2$ emissions and the installation of FGD devices in power plants (Fioletov et al., 2011; Krotkov et al., 2016). Furthermore, we estimated the trend in ground-level $SO_2$ at in-situ locations collocated with OMI. The upper panel of Fig 4 shows the trend in ground-level $SO_2$ from OMI and collocated in-situ measurements over North America for the period of 2005-2015. Both in-situ and OMI-derived ground-level $SO_2$ mixing ratios show a strong decreasing trend over eastern US mainly located at locations close to power plants. The lower panel of Fig 4 shows the scatter plot of trends in

ground-level $SO_2$ from collocated in-situ measurements and OMI. The OMI derived trends are significantly correlated (r = 0.74) with collocated in-situ trends. As expected the slope of 0.43 is similar to the absolute concentrations slope (Fig. 2) and reveals the difference in absolute trend.

Figure 5 shows the percentage change compared to 2005 in annual mean ground-level $SO_2$ concentration from coincidently-

sampled OMI and in-situ measurements and total $SO_2$ emissions from power plants over eastern US. The geographical locations of stations considered over the eastern US are shown inside the blue color box within the inset map. Both OMI and in-situ measurements shows -81±19% and -86±13% decrease in ground-level $SO_2$ over Eastern US, respectively. Earlier OMI $SO_2$ columns studies reported a 40% (Fioletov et al., 2011) and 80% (Krotkov et al., 2016) decrease near power plants in Eastern US and Ohio River Valley for the period of 2005-2010 and 2005-2015, respectively. Furthermore, we derived a -

64±18% decrease from spatially averaged OMI-derived ground-level $SO_2$ (Fig. 3) over the eastern US from entire domain (blue box in Fig. 5). The observed decrease in ground-level $SO_2$ from OMI and in-situ measurements is in agreement with the US EPA reported decrease of about 70% in total US $SO_2$ emissions (https://www3.epa.gov/airtrends/aqtrends.html).

Recently Philip et al. (2014) analyzed the $PM_{2.5}$ chemical composition over North America from the satellite data and

reported that sulfate aerosols contribute ~30% in ground-level $PM_{2.5}$ mass concentration over the eastern US. Here, the ground-level sulfate $PM_{2.5}$ mass concentration is estimated by applying the sulfate fraction from Philip et al., (2014) to the total $PM_{2.5}$ mass concentration inferred using the method of van Donkelaar et al. (2010), which uses information from satellites, models and monitors. Figure 6 shows the spatial distribution of ground-level $SO_2$ mixing ratio (left panel) and sulfate $PM_{2.5}$ mass concentration over eastern US for the period of 2005-2008. The locations of large (>18.98 kt[SO2]/yr in

2006) power plants (largest contributor to $SO_2$ emissions) and 2005-2008 average boundary-layer winds from an ECMWF (European Center for Medium range Weather Forecasting) reanalysis (Dee et al., 2011) are overlaid on the plots as circle and arrows respectively. This demonstrates that ground-level $SO_2$ influence air quality locally due to its shorter atmospheric lifetime. However, sulfate $PM_{2.5}$ with longer atmospheric lifetime influences air quality locally as well as downwind through long-range transport. It is evident from Figure 6 that column $SO_2$ and sulfate $PM_{2.5}$ hotspots are collocated around and





downwind of power plant locations. There is only a moderate spatial correlation (r = 0.60) between OMI $SO_2$ and sulfate $PM_{2.5}$ but given that sulfate is largely a secondary pollutant, this is not surprising. It was also found that there is a saturation effect at high $SO_2$ mixing ratios (Supporting Information Fig S4).

## 4 Conclusions

We examined the spatial and temporal characteristic of the ground-level $SO_2$ concentration from OMI over North America during the period from 2005-2015. OMI-derived ground-level $SO_2$ concentrations and trend correlates well with in-situ measurements (r = 0.61 and 0.74, respectively) with a significant bias in slope. Once the in-situ observations are adjusted, based on nested GEM-MACH model results, to account for the spatial sampling differences between the in-situ and OMI spatial resolution there is a notable increase (~92%) in slope to a value of 0.75. The observed reduction in ground-level $SO_2$ concentration from OMI (-81±19%) is consistent with in-situ measurements (-86±13%) over Eastern US for the period of 2005-2015. The observed decreasing trend in ground-level $SO_2$ could lead to considerable reduction in sulfate aerosols, and thus play a major role in improving air quality thereby minimizing its deleterious health impact. The long-term spatial distribution maps of ground-level $SO_2$ from OMI provide policy-makers with $SO_2$ pollution monitoring at locations where ground measurements are not available. Future satellite missions like TEMPO (Tropospheric Emissions: Monitoring Pollution) will provide better coverage of $SO_2$, and other pollutants, as it will have higher spatial resolution, and hourly frequency over the North American continent during daytime (especially USA and parts of Canada). Whereas TROPOspheric Monitoring Instrument (TROPOMI) scheduled to launch in 2016 will provide daily global coverage of tropospheric $SO_2$ and other pollutants with a high spatial resolution of 7x7 $km^2$.

## Acknowledgement

We acknowledge the National Aeronautics and Space Administration (NASA) for the availability of OMI $SO_2$ tropospheric column data. We would like to thank Natural Sciences and Engineering Research Council of Canada (NSERC) for funding support.

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



**Figure 1: Spatial distribution of mean OMI-derived ground-level SO₂ mixing ratio over North America for the period of 2005-2007, 2008-2010, 2011-2015 and 2005-2015.**





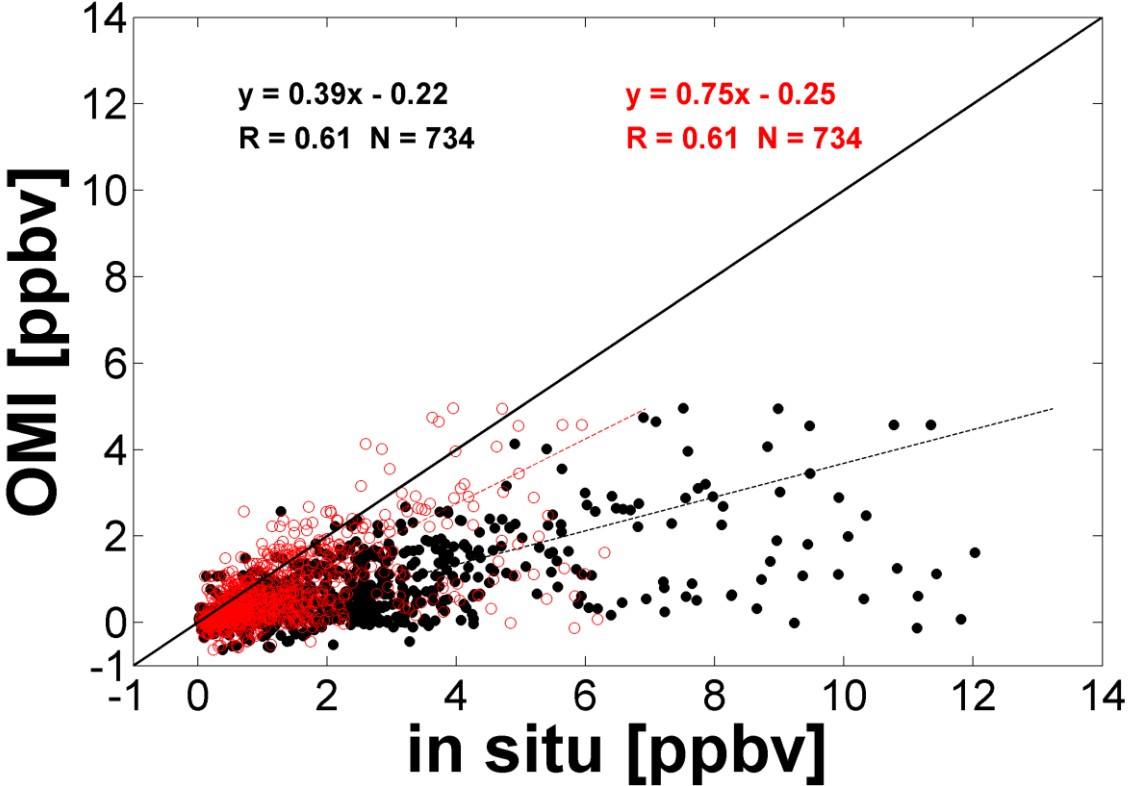

**Figure 2: Scatter plot of the annual mean OMI-derived ground-level SO₂ versus collocated in-situ measurements for the years of 2005-2015. Filled black circles represent the original in-situ values, and red circles represent the comparison with spatially inhomogeneity adjusted in-situ values.**





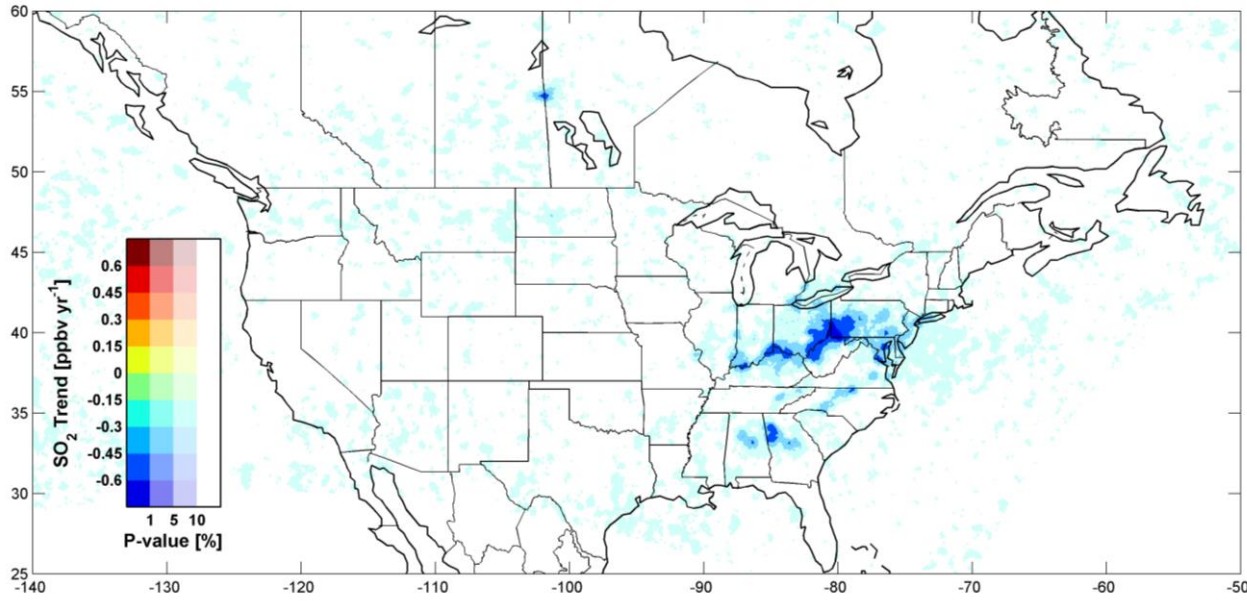

**Figure 3: Spatial distribution of OMI-derived surface SO₂ trend at 0.1° × 0.1° over North America for the year of 2005-2015. Statistical significance is shown in the form of a two sided P-value, tested against null being zero trend.**



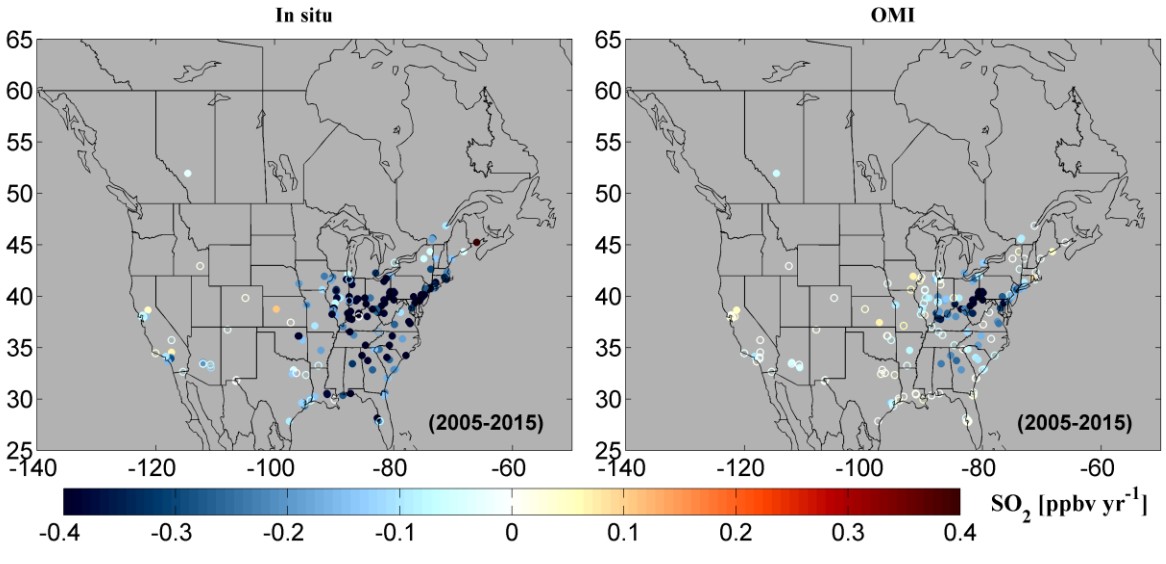

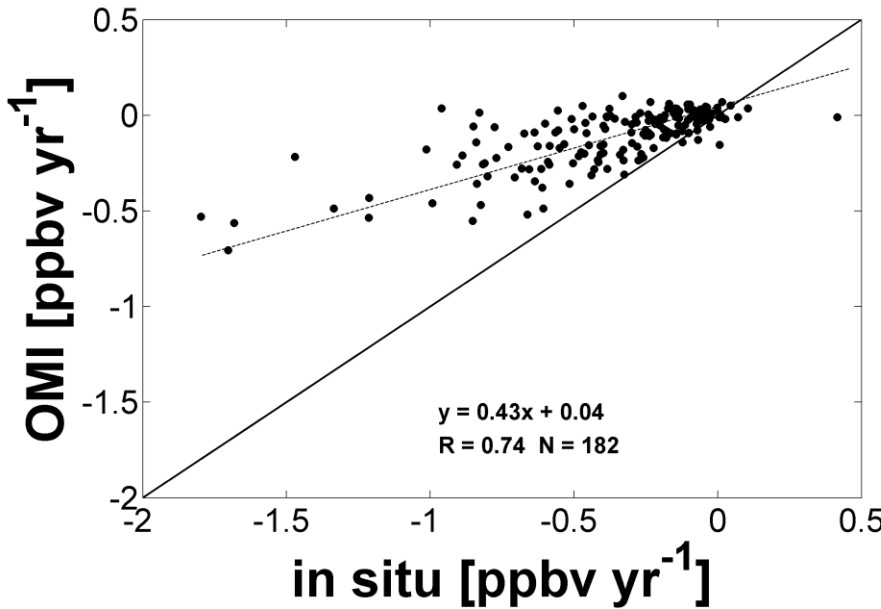

**Figure 4: Trends in ground-level SO₂ for the period of 2005-to-2015. The top row shows trends inferred from in-situ measurements at OMI overpass and from OMI for the period of 2005-2015. The filled circle represents where trend p-value < 0.05 and trend p-values > 0.05 are shown as empty circle. The bottom panel contains scatter plots of trends for the period of 2005-2015.**





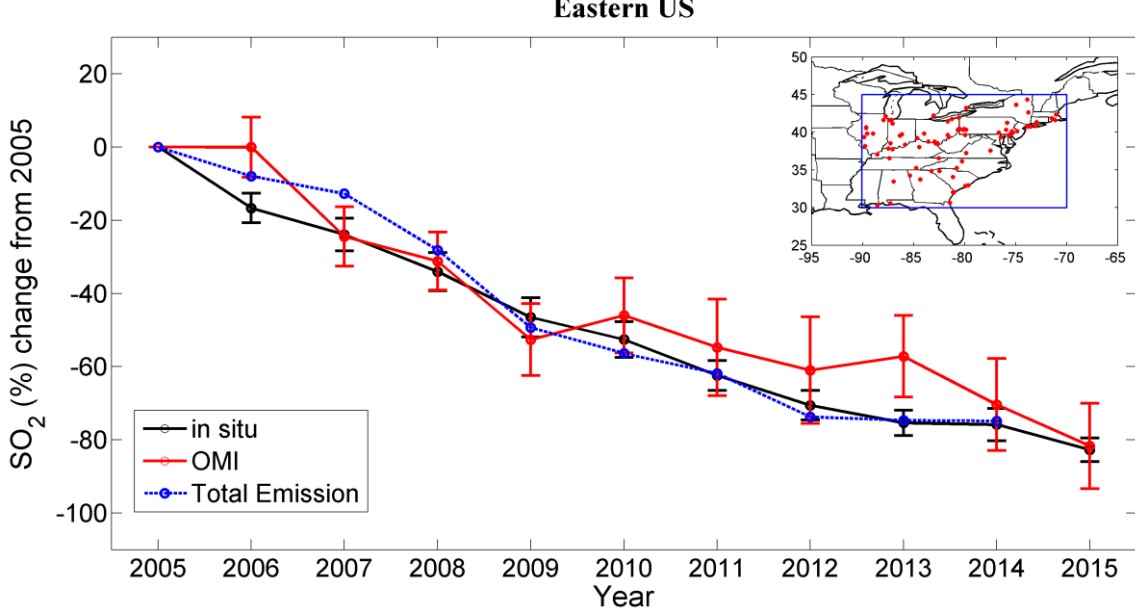

**Figure 5: Percent change in ground-level SO₂ mixing ratio from 2005 over Eastern US and southern Ontario, Canada. The in-situ and OMI ground-level SO₂ percent change are shown in black and red color, respectively. Blue circles show changes in total SO₂ emissions. The locations of in-situ measurement stations over Eastern US and southern Ontario, Canada (blue box) are shown in the inset map.**





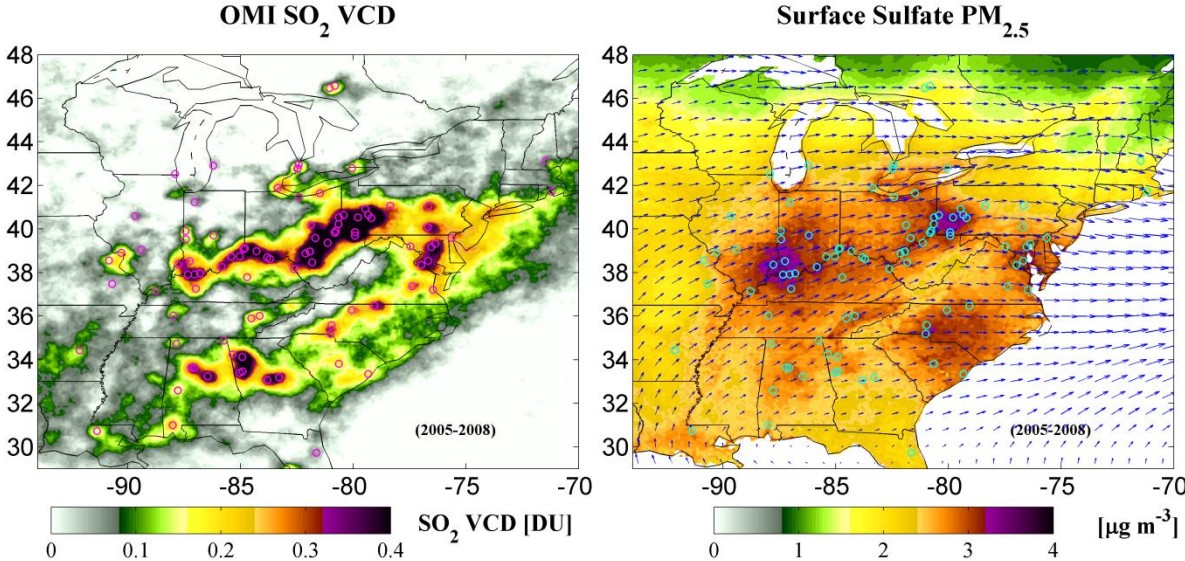

**Figure 6: Spatial distribution of satellite derived SO$_2$ vertical column density (VCD) and sulfate PM$_{2.5}$ mass concentration over eastern US and southern Ontario, Canada. The power plant locations overlaid on both panels are shown as circle. ECMWF model derived ground-level winds are overlaid on sulfate PM$_{2.5}$ mass concentration map are shown with arrows.**