# Peer review of "OMI Satellite Observations of decadal changes in Ground-Level Sulfur Dioxide over North America"

_Atmospheric Chemistry and Physics, 2016_

## Referee Comment (RC1) · Anonymous Referee #1 · 10 Jan 2017

The authors present a trend analyses of ground-level SO2 concentrations from OMI measurements over the US. This paper is interesting and has high potential, but, in my opinion, currently lacks sufficient contest and motivation. The flow of the paper is good and logical, although some sections are a bit too compact to my liking. I therefore suggest the paper to be accepted after major revisions, considering the text below.

Introduction.

The introduction clearly states the benefits of studying SO2 from satellite, mentioning its role in the formation of sulfate aerosol and the effect of the latter on climate and environmental and health issues. Related previous work is adequately cited. However, the cited paper of Krotkov et al. [2016] already gives a trend analyses of OMI total

column, over the same time period as the current paper, and furthermore indicates (for polluted areas) the close relationship between derived total columns and emissions. Although the current paper studies surface concentration rather than total column, I would like to see a more elaborate text, motivating why studying total column is not sufficient a proxy for emission trend analysis and the connected assessment of health risks. Only one short sentence is currently dedicated to the novel aspects of the paper and at first glance the overlap with previous work seems high. Please expand.

Section 2. 2.1 OMI: Concise paragraph. Line 3: 'Also, SO2 variability...' I presume background SO2 is meant here? Line 6: Please explain the use of the respective thresholds of 0.2 and 65 degree or give an reference.

2.3: Model information Line 24: It would be good to have a quantitative indication of the thickness of the lowest model layer, so reader not familiar with GEM-MACH can develop a feeling for what is assumed as 'surface concentration'. Along the same line, an indication of the partial column of the lowest layer with repect to the toal boundary layer column is missing. The reader is referred to McLinden et al papers for AMF-related information, but I think it should be discussed to some extend in the text (here or in the next paragraph).

2.4 Estimation of ground-level SO2 from OMI. I have the same problem with this paragraph as with the previous. A simple connection between observed and model concentration and column properties is given, adopted from literature, but no discussion is given. This would be ok in the case of an extende section 2.3 Line 2: The cited Lamsal [2008] paper on NO2 is missing from the list of references.

3 Results and Discussion. The actual results look sound and well described. Line 26/27: Lee et al [2011] paper is missing from the list of references. Also, this paper already derived SO2 surface mixing ratios from OMI and compared them to in-site measurements, be it only for 2006. Also this motivates a clearer description of the novel aspects of your paper in the introduction.

---

## Referee Comment (RC2) · Anonymous Referee #2 · 13 Jan 2017

In this paper, Shailesh K. Kharol et al. presented an estimation of ground-level sulfur dioxide concentrations from the Ozone Monitoring Instrument (OMI) using SO2 profiles from the Global Environmental Multi-scale – Modelling Air quality and CHemistry (GEM-MACH) model over North America for the period of 2005–2015. Also comparisons and trend analisys using OMI, GEM-MACH and in-situ SO2 observations are presented.

General observations: The paper is quite interesting but need more details. The authors should highlight their work and the novelty of this paper. SO2 from space and comparisons with ground or model observations is not a new subject, I suggest a more detailed presentation of Section 2 Data sets & methodology. The authors should show

that this paper is more than just a database manipulation.

1 Introduction

In this section the authors should connect their work with other studies, e.g. China, etc. Also, in this section, the authors should highlight the novelty of their work.

2 Data sets & methodology

In this section they should introduce more details.

3 Results & Discussion

page 5/L1:5 you should give some coordinates and to specify if Flin Flon, Snow Lake, Sudbury, Thompson, Montreal are power plants (?)

page 5/L5:10 where is apparent the closure of Flin Flon copper smelter?

Other observations: I suggest to introduce a study case using one (or more) power plants for the SO2 sources mentioned in this work.

---

## Author Comment (AC1) · 9 Mar 2017

**Point-by-point clarification to Referee #1**

In the present response letter, we summarize our modifications in the revised version of the manuscript and we provide point-by-point clarifications (in blue color) to the referee's comments and suggestions (in black color). We found the referee's comments very useful and in the right direction in order to improve the scientific quality of the paper. All of the referee's comments were taken into account in the revised version and we believe that it is now much better.

The authors present a trend analyses of ground-level $SO_2$ concentrations from OMI measurements over the US. This paper is interesting and has high potential, but, in my opinion, currently lacks sufficient contest and motivation. The flow of the paper is good and logical, although some sections are a bit too compact to my liking. I therefore suggest the paper to be accepted after major revisions, considering the text below.

We thank the referee for recommending the manuscript for publication. We have incorporated all the comments/suggestion in the revised manuscript as suggested by referee.

**Introduction.**
The introduction clearly states the benefits of studying $SO_2$ from satellite, mentioning its role in the formation of sulfate aerosol and the effect of the latter on climate and environmental and health issues. Related previous work is adequately cited. However, the cited paper of Krotkov et al. [2016] already gives a trend analyses of OMI total column, over the same time period as the current paper, and furthermore indicates (for polluted areas) the close relationship between derived total columns and emissions. Although the current paper studies surface concentration rather than total column, I would like to see a more elaborate text, motivating why studying total column is not sufficient a proxy for emission trend analysis and the connected assessment of health risks. Only one short sentence is currently dedicated to the novel aspects of the paper and at first glance the overlap with previous work seems high. Please expand.

We agree with referee's view point and included more detail on novel aspects of the paper in the revised manuscript. The total column $SO_2$ is a sufficient proxy for emission trend analysis but is of low utility for the assessment of health risks. We include the sentence "In contrast to total column $SO_2$, long-term records of ground-level $SO_2$ concentrations from satellite observations will be directly useful to assess air quality and associated health risks." in the revised manuscript at Page number 2, Line number 18-20.

Section 2. 2.1 OMI: Concise paragraph. Line 3: 'Also, SO2 variability...' I presume background SO2 is meant here? Line 6: Please explain the use of the respective thresholds of 0.2 and 65 degree or give an reference.

We have modified the sentence as "Also, $SO_2$ retrieval variability is reduced by a factor of two in the PCA algorithm relative to the BRD algorithm (Li et al., 2013)" at Page number 3, Line number 8-9, and provided reference for respective thresholds at Page number 3, Line number 18 in the revised manuscript.

2.3: Model information Line 24: It would be good to have a quantitative indication of the thickness of the lowest model layer, so reader not familiar with GEM-MACH can develop a feeling for what is assumed as 'surface concentration'. Along the same line, an indication of the partial column of the lowest layer with respect to the total boundary layer column is missing. The reader is referred to McLinden et al papers for AMF related information, but I think it should be discussed to some extend in the text (here or in the next paragraph).

We have modified the sentence as "The lowest model layer, which is 20 m thick, is taken as ground-level concentration." in the revised manuscript at Page number 4, Line number 11-12.

We have included the following paragraph on AMF related information in section 2.1 at Page number 3, Line number 9-16:-

"Even though the PCA algorithm directly estimates $SO_2$ vertical column density in one step using $SO_2$ Jacobians, the air mass factor (AMF) is effectively fixed at 0.36 (representing summertime conditions in the eastern USA), similar to the BRD algorithm. A better estimation of AMFs is needed for different regions to reduce these systematic errors that result from conditions that do not match these. For this, we re-calculated the AMFs using $SO_2$ profile information from the high resolution (15 km x 15 km) GEM-MACH air quality forecast model (discussed in section 2.3), monthly-varying surface reflectivity from the MODIS satellite instruments, and an improved identification of snow. More details on Environment Canada Air Mass Factors calculation for $SO_2$ are discussed in McLinden et al., 2014; 2016b."

2.4 Estimation of ground-level $SO_2$ from OMI. I have the same problem with this paragraph as with the previous. A simple connection between observed and model concentration and column properties is given, adopted from literature, but no discussion is given. This would be ok in the case of an extended section 2.3 Line 2: The cited Lamsal [2008] paper on $NO_2$ is missing from the list of references.

We have included Lamsal [2008] reference and more detail on AMF calculation in section 2.1in the revised manuscript.

"Lamsal, L. N., Martin, R. V., van Donkelaar, A., Steinbacher, M., Celarier, E. A., Bucsela, E., Dunlea, E. J., and Pinto, J. P.: Ground-level nitrogen dioxide concentrations inferred from the satellite-borne Ozone Monitoring Instrument, J. Geophys. Res., 113, D16308, doi:10.1029/2007JD009235, 2008."

3 Results and Discussion. The actual results look sound and well described. Line 26/27: Lee et al [2011] paper is missing from the list of references. Also, this paper already derived $SO_2$ surface mixing ratios from OMI and compared them to in-site measurements, be it only for 2006. Also this motivates a clearer description of the novel aspects of your paper in the introduction.

We have included Lee et al [2011] reference in the revised manuscript.

"Lee, C., Martin, R. V., van Donkelaar, A., Lee, H., Dickerson, R. R., Hains, J. C., Krotkov, N., Richter, A., Vinnikov, K., and Schwab, J. J.: $SO_2$ emissions and lifetimes: Estimates from

inverse modeling using in situ and global, space-based (SCIAMACHY and OMI) observations, J. Geophys. Res., 116, D06304, doi:10.1029/2010JD014758, 2011."

We have included the following sentence in the introduction at Page number 2, Line number 16-17.

"In previous studies (Lee et al., 2011; Nowlan et al., 2011), ground-level $SO_2$ concentrations were estimated for only a one year period using satellite observations over North America."

---

## Author Comment (AC2) · 9 Mar 2017

**Point-by-point clarification to Referee #2**

We thank the referee for his valuable comments/suggestions. We found the referee's comments very useful and in the right direction in order to improve the scientific quality of the paper. The point-by-point clarifications (in blue color) to the referee's comments and suggestions (in black color) are given below.

In this paper, Shailesh K. Kharol et al. presented an estimation of ground-level sulfur dioxide concentrations from the Ozone Monitoring Instrument (OMI) using $SO_2$ profiles from the Global Environmental Multi-scale – Modelling Air quality and Chemistry (GEM-MACH) model over North America for the period of 2005–2015. Also comparisons and trend analysis using OMI, GEM-MACH and in-situ $SO_2$ observations are presented.

General observations: The paper is quite interesting but need more details. The authors should highlight their work and the novelty of this paper. $SO_2$ from space and comparisons with ground or model observations is not a new subject, I suggest a more detailed presentation of Section 2 Data sets & methodology. The authors should show that this paper is more than just a database manipulation.

We have incorporated all the comments/suggestion in the revised manuscript as suggested by the referee. The point-by-point clarifications to the referee's comments are as follows:

1 Introduction
In this section the authors should connect their work with other studies, e.g. China, etc. Also, in this section, the authors should highlight the novelty of their work.

We agree with referee's view point and included the sentence "In contrast to total column $SO_2$, long-term records of ground-level $SO_2$ concentrations from satellite observations will be directly useful to assess air quality and associated health risks" in revised manuscript at Page number 2, Line number 18-20 to highlight the novel aspects of the paper.

The cited Fioletov et al., (2013) and Krotkov et al., (2016) papers are examples of related studies over China and other regions.

2 Data sets & methodology
In this section they should introduce more details.

We have included more details in data sets & methodology section in the revised manuscript.

The following paragraph on AMF related information is included in section 2.1 at Page number 3, Line number 9-16:-

"Even though the PCA algorithm directly estimates $SO_2$ vertical column density in one step using $SO_2$ Jacobians, the air mass factor (AMF) is effectively fixed at 0.36 (representing summertime conditions in the eastern USA), similar to the BRD algorithm. A better estimation of AMFs is needed for different regions to reduce these systematic errors that result from

conditions that do not match these. For this, we re-calculated the AMFs using $SO_2$ profile information from the high resolution (15 km x 15 km) GEM-MACH air quality forecast model (discussed in section 2.3), monthly-varying surface reflectivity from the MODIS satellite instruments, and an improved identification of snow. More details on Environment Canada Air Mass Factors calculation for $SO_2$ are discussed in McLinden et al., 2014; 2016b."

3 Results & Discussion
page 5/L1:5 you should give some coordinates and to specify if Flin Flon, Snow Lake, Sudbury, Thompson, Montreal are power plants (?)

We have modified the sentence as "There are far fewer sources in the western US and Canada, with a few notable exceptions such as Flin Flon (54.77$^o$ N, 101.88$^o$ W; copper smelter), Sudbury (46.52$^o$ N, 80.95$^o$ W; copper and nickel smelter), Thompson (55.74$^o$ N, 97.85$^o$ W; metal ore mining), Montreal (45.50$^o$ N, 73.56$^o$ W), the oil sands region in northern Alberta and power plants nearby Edmonton." in the revised manuscript at Page number 5, Line number 14-17.

page 5/L5:10 where is apparent the closure of Flin Flon copper smelter?

We have modified the sentence as "The closure of Flin Flon (54.77$^o$ N, 101.88$^o$ W) copper smelter in June 2010 is also apparent in OMI-derived ground-level $SO_2$ during 2011-2015 in Figure 1." in the revised manuscript at Page number 5, Line number 21-22.

Other observations: I suggest to introduce a study case using one (or more) power plants for the $SO_2$ sources mentioned in this work.

We have included following time series figure for Bowen power plant (34.13$^o$ N, 84.92$^o$ W), USA, and Flin Flon copper smelter (54.77$^o$ N, 101.88$^o$ W), Canada in the revised manuscript.

The following figure description is included at Page number 7, Line number 6-9 for the figure below in the revised manuscript.

[Figure]

Figure 6 shows that bottom-up $SO_2$ emissions and OMI-derived ground-level $SO_2$ concentrations are temporally correlated even for larger individual point sources, namely the Bowen power plant ($34.13^{o}$ N, $84.92^{o}$ W), USA, and Flin Flon copper smelter ($54.77^{o}$ N, $101.88^{o}$ W), Canada. The bottom-up emission data for these sites are obtained from the US EPA (2016), and National Pollutant Release Inventory (NPRI, 2017), respectively.